# Adhesion and Activation of Blood Platelets on Laser-Structured Surfaces of Biomedical Metal Alloys

**DOI:** 10.3390/jfb14090478

**Published:** 2023-09-18

**Authors:** Marta Kamińska, Aleksandra Jastrzębska, Magdalena Walkowiak-Przybyło, Marta Walczyńska, Piotr Komorowski, Bogdan Walkowiak

**Affiliations:** 1Department of Biophysics, Institute of Materials Science and Engineering, Lodz University of Technology, Stefanowskiego 1/15, 90-537 Lodz, Poland; marta.kaminska@p.lodz.pl (M.K.);; 2Aflofarm Farmacja Polska Sp. z o.o., 95-200 Pabianice, Poland; 3Department of Medical Imaging Techniques, Medical University of Lodz, Lindley 6, 90-131 Lodz, Poland; 4Bionanopark Laboratories, Bionanopark Sp. z o.o., Dubois 114/116, 93-465 Lodz, Poland

**Keywords:** metallic biomaterials, laser surface modification, platelet adhesion, platelet activation, platelet aggregation

## Abstract

The laser surface modification of metallic implants presents a promising alternative to other surface modification techniques. A total of four alloyed metallic biomaterials were used for this study: medical steel (AISI 316L), cobalt–chromium–molybdenum alloy (CoCrMo) and titanium alloys (Ti6Al4V and Ti6Al7Nb). Samples of metallic biomaterials after machining were subjected to polishing or laser modification in two different versions. The results of surface modification were documented using SEM imaging and roughness measurement. After modification, the samples were sterilized with dry hot air, then exposed to citrate blood, washed with PBS buffer, fixed with glutaraldehyde, sputtered with a layer of gold and imaged using SEM to enable the quantification of adhered, activated and aggregated platelets on the surface of biomaterial samples. The average total number, counted in the field of view, of adhered platelets on the surfaces of the four tested biomaterials, regardless of the type of modification, did not differ statistically significantly (66 ± 81, 67 ± 75, 61 ± 70 and 57 ± 61 for AISI 316L, CoCrMo, Ti6Al4V and Ti6Al7Nb, respectively) and the average number of platelet aggregates was statistically significantly higher (*p* < 0.01) on the surfaces of AISI 316L medical steel (42 ± 53) and of the CoCrMo alloy (42 ± 52) compared to the surfaces of the titanium alloys Ti6Al4V (33 ± 39) and Ti6Al7Nb (32 ± 37). Remaining blood after contact was used to assess spontaneous platelet activation and aggregation in whole blood by flow cytometry. An in-depth analysis conducted on the obtained results as a function of the type of modification indicates small but statistically significant differences in the interaction of platelets with the tested surfaces of metallic biomaterials.

## 1. Introduction

Metallic biomaterials, mainly metal alloys, remain the basic group of structural biomaterials for bone implants in orthopaedics, spinal surgery and dentistry. The obvious reason for the widespread popularity of metallic implants is their good strength properties, as well as a good level of tolerance by the human body [1]. However, efforts are being made to modify these biomaterials while maintaining their strength properties so as to achieve a better degree of osseointegration while improving the biocompatibility of these surfaces [2] and their resistance to microbial colonization [3]. In order to achieve the assumed goals, various strategies of surface modification of these biomaterials are adopted. A typical and very popular method is to cover the surface of the implant with a layer of hydroxyapatite, either by spraying or by synthesis of the layer on the surface of the implant [4]. Spraying a layer of porous spongy titanium [5] or electrochemically coating the surface of a titanium implant with titanium nanotubes [6] or porous titanium [7] is gaining similar popularity. In addition, attempts are being made to deposit carbon layers, for example in the form of nanocrystalline diamond-like carbon [8]. In many cases, attempts are made to use mixed techniques to obtain hybrid layers, often enriched with other chemical elements [9]. A different approach is the production of metal implant structures by 3D printing [10] and laser modification of metallic surfaces via an ablation process and local surface re-melting of the material [11]. The latter strategy of metallic surface modification is the subject of our current work, the aim of which is the selection of the optimal metallic material and modification of its surface with the use of a laser to obtain optimal results in the design and manufacture of orthopaedic bone implants for effective direct percutaneous fixation of the knee and feet prosthesis to the bone in patients following above-the-knee amputation. This technique of percutaneous prosthesis fixation (intraosseous transcutaneous amputation prosthesis—ITAP) relieves the soft tissues from carrying the loads that occur in the case of classic prosthesis fixation with the use of an orthopaedic socket [12]. This technique is very popular in dental prosthetics, where the implant is implanted directly into the bone, and its other end protrudes above the mucous membranes and is the place where the tooth crown is attached [13]. It is believed that the structure of the implant surface (topography) is essential for proper osseointegration, enabling permanent anchoring of the bone tissue on the structured surface. Hence the desire to create a porous surface with appropriate roughness (hydroxyapatite, porous spongy titanium, openwork structure of the implant, laser ablation and melting of the material). The surface of the implant is the site of complex biological processes during contact with the body’s tissues [14], including processes of interaction with the blood and influence on the body’s immunity [15]. On the other hand, such surface modifications usually produce a favourable environment for the formation of a microbial biofilm. Therefore, the susceptibility of the resulting surface to microbial colonization must be carefully determined, which can eliminate even the most promising surface modifications from reaching practical application. Another parameter that should be taken into account is the susceptibility of the surface to the adhesion of proteins from body fluids, which facilitates the formation of a protein matrix that promotes cell adhesion and the process of osseointegration itself. The adhesion of platelets from damaged blood vessels of the surrounding tissues to the implant surface is also not to be missed. Adhered and activated platelets are a very rich source of growth factors that promote the proliferation of remodelling bone tissue [16].

Our earlier reports [17,18] were devoted to the optimization of the conditions for laser modification of the surface of titanium alloy Ti6Al4V in order to obtain a satisfactory proliferation of bone cells on the modified surface while limiting the susceptibility of this surface to microbial colonization.

The aim of this report was to assess the susceptibility to platelet adhesion, activation and aggregation on the surface of four different popular metallic biomaterials in the form of metal alloys, subjected to four different surface modification treatments. Additionally, the distant in time and space activation and aggregation of platelets in whole anticoagulated blood after contact with the tested surfaces were assessed. The latter may be important in determining the risk of deep vascular thrombosis resulting from insertion of the implant during surgical intervention. 

## 2. Materials and Methods

### 2.1. Metal Alloys

Tests were carried out for medical alloys: medical steel AISI 316L—(UGITECH ITALIA (Peschiera Borromeo, MI, Italy), ASTM F138-13/ISO5832.1 2016), cobalt-chromium-molybdenum alloy CoCrMo—(CARPENTER TECHNOLOGY, Philadelphia, PA, USA, ASTM F1537-11), titanium–aluminium–vanadium alloy Ti6Al4V ELI—(VSMPO-AVISMA, Verkhnyaya Salda, Russia, ASTM F136-08e1/ISO 5832-3 1996), titanium–aluminium–niobium alloy Ti6Al7Nb—(WOLFTEN, Wroclaw, Poland, ASTM F1295/ISO 5832-11).

### 2.2. Chemicals

Fluorescently labelled CD62-PE, CD61-perCP antibodies and CellFix reagent were purchased from Becton Dickinson (Franklin Lakes, NJ, USA), glutaraldehyde, absolute ethyl alcohol, phosphate buffered saline, pH 7.4 (PBS), adenosine diphosphate (ADP) and sodium citrate were purchased from Sigma Aldrich (Saint Louis, MI, USA).

### 2.3. Preparation of Test Samples

Test samples were prepared in accordance with the requirements of ISO 10993-12:2021. The initial materials in the form of rods with a diameter of 8 mm were cut into 3 mm thick discs, which were then modified:

Type A modification—mechanical grinding

Type B modification—mechanical grinding/polishing

Type C modification—laser modification (rosette-like effect)

Type D modification—laser modification (crater-like effect)

The samples previously ground (modification type A) were subjected to laser modification using Laser Da Vinci 1300 (C.B. Ferrari, Modena, Italy). The laser operating parameters are presented in Table 1, while a diagram of the laser-modified sample is shown in Figure 1. In addition, a sample of polished medical steel was used in all experiments, acting as an internal laboratory standard allowing for a quantitative comparison of the results obtained in separate experiments and checking the correctness of the experiment.

The samples prepared in this way were washed and sonicated in 70% ethanol (15 min) and then washed and sonicated twice in distilled water (15 min). The dried samples were sterilized with hot dry air using the SRW 115 STD sterilizer (POL-EKO Apparatus, Wodzislaw Slaski, Poland) at 180 °C for 45 min.

### 2.4. Microscopic Observations of Sample Surfaces

The surfaces of the prepared samples were subjected to microscopic observation both before contact with blood and after the incubation period. A scanning electron microscope (SEM) JSM-6610LV (JEOL, Tokyo, Japan) was used for microscopic observations. After contact with blood, surfaces of the samples were covered with a thin (6 nm) layer of gold using a Leyca ACE6000 sputtering machine (Leica Microsystems, Buffalo Grove, IL, USA). For each of the analysed samples, 10 photos were taken in randomly selected places, with a magnification of 1000×.

### 2.5. Assessment of Surface Roughness of Samples

The roughness of the samples was assessed using a HOMMEL TESTER T1000 contact profilometer equipped with the EVOVIS 2.00.1.00 software (JENAOPTIK Industrial Metrology, Jena, Germany). Surface tests were carried out using a scanning speed of 0.15 mm/s and a measurement range of 400 μm with a distance between points of 0.4 μm and a mapping distance of 4 mm. On each characterized surface, five measurements were made along randomly selected directions. The results of the arithmetic average of profile height deviations from the mean line parameter (Ra) are presented in the form of mean values (MEAN) and the corresponding values of standard deviations (SD). In addition, roughness evaluation was carried out using an S neox non-contact optical profilometer (Sensofar, Spain) equipped with an EPI 20X v35 lens. The sample was tested on a surface of 850.08 × 709.32 μm, and the analysis of roughness parameters was performed using the SensoSCAN 6 software. The measurement results are presented as the average value of the Sa parameter (arithmetical mean height) calculated for the tested surface.

### 2.6. Blood Collection

Blood for testing was collected immediately before the experiment (approximately 20–30 min) from healthy volunteers who had not taken antiplatelet drugs for 2 weeks prior to donation. Blood was collected, via single puncture into a vein, on the anticoagulant—sodium citrate (0.1 mol/L). The research was carried out with the consent of the Local Ethics Committee at the Medical University of Lodz (RNN/160/03/KE) granted to conduct research related to the interaction of biomaterials with human blood. All blood experiments were performed in at least three independent replicates.

### 2.7. Contact of Blood with the Tested Surface

Sterile samples were placed in Eppendorf tubes (1.5 mL) and exposed to citrated whole blood for one hour at 37 °C and under gentle flow conditions (end to end mixing). The samples were then removed from the tubes and rinsed three times with PBS, pH 7.4 (37 °C), and fixed with 2.5% glutaraldehyde solution (one hour). After fixation, the samples were again rinsed with PBS buffer and dehydrated in ethyl alcohol of increasing concentration (50%, 70%, 80%, 90%, absolute, 10 min each). Three samples were analysed simultaneously for each of the tested biomaterials. Subsequently, ten surface images obtained via SEM were archived, on which counts of the number of individual platelets and platelet aggregates were made. The obtained results are presented in the tables in the form of two parameter values: the number of platelets in particular degrees of activation according to the Goodman scale [19] and the number of platelet aggregates. To simplify the analysis, Goodman’s five-point scale was modified to three levels: no activation (round platelets), moderate activation (plates with formed pseudopodia), and strong activation (plates fully spread over the surface). Meanwhile, platelet aggregates were divided into two groups: small (up to 10 plates) and large platelet aggregates. Each type of biomaterial in each modification was tested in four independent experiments using three samples in each of the experiments, giving a total of 120 images for each of the tested samples. The blood remaining in the test tube was used to study the spontaneous activation and aggregation of platelets after contact with the tested surface.

### 2.8. Evaluation of Spontaneous Platelet Activation and Aggregation in Whole Blood after Contact with the Test Surface

Blood samples previously subjected to one hour contact with the tested biomaterials were used for the study. Analysis of platelet activation and aggregation in citrated whole blood was performed using an Accuri C6 flow cytometer (Becton Dickinson, Franklin Lakes, NJ, USA). Each experiment was independently repeated four times with three replicate samples in each experiment. The specific antibody against β3 integrin CD61-perCP was used to identify platelets. To assess the level of platelet activation after contact with the tested surfaces, a specific marker of platelet activation was used—Selectin P (CD62 P)—exposed on the surface of platelets as a result of reorganization of the platelet membrane of the activated platelet [20]. In our experiment, the CD62 antibody conjugated with phycoerythrin (CD62 PE) specifically recognizing P-Selectin on the surface of activated platelets was used [21]. Spontaneous platelet aggregation was assessed by SSC/FSC dot cytometric plots (SSC—side scattered laser light, FSC—forward scattered laser light) [22,23]. After contact with the tested biomaterials, blood samples were fixed using CellFix reagent. The negative control was a blood sample without contact with the tested biomaterials, and the positive control was a blood sample to which ADP was added at the final concentration of 10 µM (without re-calcination). The amounts of antibodies and CellFix reagent added were as recommended by the manufacturer.

### 2.9. Statistical Evaluation

Statistical analysis of the results was performed using one-way ANOVA. It was assumed that if the value of statistical significance *p* was less than 0.05, then the results differ statistically significantly. Statistical significance levels are marked with asterisks as follows: * *p* < 0.05, ** *p* < 0.01, *** *p* < 0.001.

## 3. Results

### 3.1. Imaging of Tested Surfaces before Contact with Blood

The surfaces of metallic biomaterials prepared for testing in contact with blood were observed using SEM microscopy. Figure 2 and Figure 3 present representative images of the surfaces of the tested samples at low (30×) and high (1000×) magnification.

### 3.2. Roughness Analysis of the Tested Surfaces

The surfaces of the samples prepared for testing were subjected to roughness analysis. Values of the parameter Ra (arithmetic average of profile height deviations from the mean line) obtained using the HOMMEL TESTER T1000 contact profilometer for five randomly selected measurement lines and the values of the parameter Sa (arithmetic mean of height calculated for the analysed surface) obtained using the non-contact optical profilometer S neox are summarized in Table 2.

Sample images of the surface of the tested samples, imaged with the S neox non-contact optical profilometer, are shown in Figure 4.

### 3.3. Imaging of Tested Surfaces after Contact with Blood

After contact with blood, the examined surfaces were re-imaged using SEM. A total of 120 images were analysed for each type of biomaterial and each modification. Adhered platelets, both single and in the form of aggregates of various sizes, were visible. Numerous leukocytes (large, bright spherical objects) and erythrocytes (large objects with a characteristic concavity) trapped on the surface of the examined materials were also visible. Figure 5 shows representative images of sample surfaces after contact with blood.

The analysis of the obtained images allowed us to numerically characterize the level of activation (Table 3) and aggregation (Table 4) of platelets adhered to the surface of the tested biomaterials. For each type of biomaterial and for each surface modification, 120 images were analysed, and the numerical results obtained, relating to the field of view area, were grouped as described in the Section 2.7. Contact of Blood with the Tested Surface.

### 3.4. Activation of Platelets on the Tested Surfaces

Table 3 presents and compares the number of platelets adhered to the tested surfaces, broken down by the type of material and its surface modification, as well as the level of activation of the adhered platelets. The last column shows statistical significance of differences in the average number of platelets per biomaterial depending on its surface modification and activation level. The notation “*** A–D, B–D, C–D”, located in the first row of Table 3, means a statistically significant difference, at the level of *p* < 0.001, between values of the numbers of adhered, but not activated, platelets on the surface of AISI 316L medical steel for modification A compared to modification D, as well as modifications B and C compared to modification D. The column labelled “Mean per biomaterial” collects the average values of the number of platelets found on a given bio-material, regardless of the type of surface modification and the level of activation (left sub-column), and the average values of the number of platelets found on a given biomaterial, regardless of the type of surface modification, but broken down by activation level (right sub-column). The last line shows statistical significance of differences in the average number of platelets per type of modification, depending on the type of biomaterial and the level of platelet activation. The notation “* {1}–{4}, {2}–{4}” contained in the first row of the right sub-column described as “Mean per biomaterial” means a statistically significant difference, at the level of *p* < 0.05, between values of the numbers of adhered non-activated platelets on steel AISI 316L {1} and Ti6Al7Nb titanium alloy {4}and on CoCrMo alloy {2} and Ti6Al7Nb alloy {4}, without taking into account the type of surface modification. The notation “ns” means no statistically significant differences within the compared groups.

### 3.5. Platelet Aggregation on the Tested Surfaces

Table 4 presents and compares the number of platelets adhered to the tested surfaces, broken down by their surface modification and the level of aggregation of the adhered platelets. Table 4 is organized analogously to Table 3 except that the level of platelet activation has been replaced by the level of platelet aggregation.

### 3.6. Spontaneous Activation of Platelets in Whole Blood after Contact with the Surfaces of the Tested Biomaterials

Table 5 summarizes the results of the evaluation of spontaneous platelet activation in citrated whole blood after contact with the surfaces of the tested biomaterials. Platelet activation was assessed on the basis of platelet activation markers, i.e., P-Selectin (CD62P), recognized by the CD62-PE antibody. The last column and the last row contain information on statistically significant differences between individual pairs within the analysed groups. The entries “* C–D” and “*** A–D, B–D” in the third row from the top of Table 5 mean that for the CoCrMo alloy a statistically significant difference in P-Selectin expression was found between modification C and D at the level of *p* < 0.05, as well as a statistically significant difference between modifications A and D and B and D at the level of *p* < 0.001. Where statistically significant differences were not observed, the notation “ns” is used. Statistically significant differences at *p* < 0.001 were also observed for marker of platelet activation between negative and positive controls.

### 3.7. Spontaneous Aggregation of Platelets in Whole Blood after Contact with the Surfaces of the Tested Biomaterials

In turn, Table 6, organized similarly to Table 5, contains information on spontaneous platelet aggregation, expressed as the percentage of platelets in the form of aggregates. In this case, no statistically significant differences in the level of platelet aggregation after contact with the tested biomaterial surfaces were observed, while a statistically significant difference was observed at the level of *p* < 0.001 between positive and negative controls.

## 4. Discussion

Modification of the implant surface is considered as a far-reaching alternative in the search for new biomaterials with the expected physicochemical properties and acceptable biological responses of the organism in contact with these biomaterials. One of the promising methods of surface modification of metallic implants is laser structuring of the implant surface [24]. As a result of the ablation process and local surface metal re-melting, it allows for a relatively large possibility of obtaining various surface properties even within one piece of biomaterial, practically with no limits due to the shape of this piece. Our previous experience in this area has already been reported [17,18,25], but we have not yet presented full data on the assessment of thrombogenicity of surfaces modified in this way.

The samples prepared for testing were subjected to surface topography evaluation using SEM imaging and roughness measurements using dedicated devices for measuring surface roughness, both contact and optical profilometers. The microscopic analysis (Figure 2 and Figure 3) clearly shows the variation in surface topography resulting from the method of its preparation. When a laser is used, a clear geometric pattern is visible consisting of the modified area (rosettes or craters) and the area as before the laser modification. In Figure 2 and Figure 3, in column C, the differences in the structures of the modified surfaces are clearly visible, where there is some similarity between these structures for pairs of alloys AISI 316L and CoCrMo and Ti6Al4V and Ti6Al7Nb. Similarities in pairs and differences between pairs result from differences in melting temperatures of these alloys. AISI 316L steel melts at about 1375 °C [26], and CoCrMo alloy at 1290 °C [27], while titanium alloys have much higher melting points, Ti6Al4V—1649 °C [28] and Ti6Al7Nb—1720 °C [29]. Column D shows crater-like structures practically the same for all investigated metal alloys. The differences in the topography of the laser-modified surfaces result from the differences in the parameters of the processes carried out and indicate the possibility of obtaining a large variety of surface modifications. It is worth noting that the same, for all tested biomaterials, surface treatment conditions by machining or grinding do not significantly affect the differences in surface roughness of these tested biomaterials. On the other hand, the C-type (rosette-like) laser modification, always carried out with the same process parameters, leads to statistically different Ra parameter values, with the highest Ra values for titanium alloys. The D-type modification (crater-like) caused statistically insignificant differences in the Ra parameter for the tested biomaterials, but the differences in the Sa parameter are clearly visible. However, within each of the tested materials, the various types of surface modifications led to statistically significant differences in the Ra parameter, which was also reflected in the values of the Sa parameter.

The samples prepared and characterized in this way were brought into contact with blood, and as a result, the adhesion of morphotic elements to the surfaces of the tested samples was observed (Figure 5). Lymphocytes and erythrocytes can be identified there, but platelets in various stages of activation and aggregation are the most numerous. In the column marked “Mean per biomaterial” in Table 3, the total number of platelets found in the field of view on the surfaces of individual biomaterials was collected, without differentiation due to the type of modification, and this number did not differ statistically significantly. However, a deeper analysis, broken down by platelet activation levels, reveals the existence of statistically significant differences. In all tested samples, platelets with a moderate activation level are the most abundant and non-activated platelets are the least. A similar analysis carried out within individual modifications showed the greatest variation in platelet count in strong activation between CoCrMo and Ti6Al7Nb alloys, especially for the D-type modification. For the same modification, statistically significant differences in platelet count in moderate activation were also observed between AISI 316L medical steel and both titanium alloys. The analysis of platelet activation depending on the type of modification within each of the biomaterials shows the existence of statistically significant differences in the analysed groups of platelet activation, except for strong activation on the CoCrMo alloy and moderate activation on the Ti6Al7Nb alloy, where the differences were only marginally significant.

However, a similar analysis of platelet aggregation (Table 4), categorized as single platelets, small platelet aggregates (up to 10 platelets) and large platelet aggregates, shows a statistically significant difference in the total number of platelets in aggregates with a significantly higher number of individual platelets and small aggregates found on AISI 316L and CoCrMo surfaces compared to Ti6Al4V and Ti6Al7Nb titanium alloys. For large aggregates, the differences were not statistically significant. In the case of type A modification, the highest number of single platelets was found on the CoCrMo alloy and the difference was significant compared to other surfaces. Differences in the number of single platelets were also observed for the remaining modifications, except for type C, where in the case of modification B, the lowest number of single platelets was on the surface of the Ti6Al4V alloy, and in the case of modification D, the lowest number of such platelets was on the Ti6Al7Nb alloy, and the increase in number of single platelets on the tested biomaterials was in the order: Ti6Al4V, CoCrMo, AISI 316. The analysis carried out within each of the biomaterials and their surface modifications also showed significant differences with the exception of the Ti6Al7Nb alloy and single platelets. It is worth noting that the highest numbers of aggregates were found for each of the biomaterials on the surface with the D (crater-like) modification.

Only a limited number of platelets present in whole blood adhere to the surface of the biomaterial. The vast majority of platelets, despite contact with the tested surface, remain in suspension, but the fate of these platelets may depend on their contact with these surfaces. Platelets may undergo spontaneous activation and aggregation, and this process may differ from the spontaneous process in control blood where contact with the surface of the biomaterial was prevented. In other words, platelets can remember the history of contact with a surface, which can translate into their different reactivity and, as a result, may promote thrombosis or bleeding diathesis. Table 5 summarizes the results of spontaneous platelet activation upon contact with the test surface. Platelet activation marker P-Selectin was used to assess platelet activation. No statistically significant differences in platelet activation after contact with individual biomaterials were observed within individual modifications; however, differences were observed within individual biomaterials for various types of modifications, except for the Ti6Al4V alloy.

However, there were no statistically significant differences in spontaneous platelet aggregation after contact with the tested surfaces (Table 6) assessed on the basis of laser light scattering on the analysed platelets and platelet aggregates, but this may be due to the use of sodium citrate as an anticoagulant and no re-calcination.

We encountered great difficulty when trying to compare our results with reports available in the literature. The reason for this is that the authors of these reports adopted different research strategies and used different materials. The review paper [30] collected information on haemocompatibility, including aspects of platelet activation in contact with various biomaterials in vitro, while the paper [31] examined only commercial titanium platelets in in vitro contact with citrate platelet-rich plasma or suspension of isolated platelets. Although in the work [32] the in vitro adhesion of platelets to the same set of materials as in our work was analysed, the conditions of the experiment differed vastly from those used by us, because the surfaces of the samples were contacted with platelet-rich plasma, and not with whole blood, and for only 5 min. The authors presented in the report representative images of the surface of metallic samples with visible adhered platelets, but as a result they provided only the ranking of the susceptibility of these alloys to platelet adhesion, which does not allow for comparison with the numerical results and statistical analysis presented herein.

## 5. Conclusions

The results presented in this paper indicate the possibility of obtaining very subtle but statistically significant differences in the biological reaction of the modified surfaces to contact with blood. This allows the selection of appropriate conditions for laser surface modification depending on the expected contact conditions of the implant with the surrounding tissue environment. In conjunction with the previously published results, it also allows for the thesis that laser modification of the surface of metallic biomaterials enables the design and production of an implant surface with the expected properties in contact with tissues while satisfactorily maintaining its mechanical properties. A particularly interesting feature of laser surface modification is the ability to obtain surface zones with varied but well-controlled properties.

## Figures and Tables

**Figure 1 jfb-14-00478-f001:**
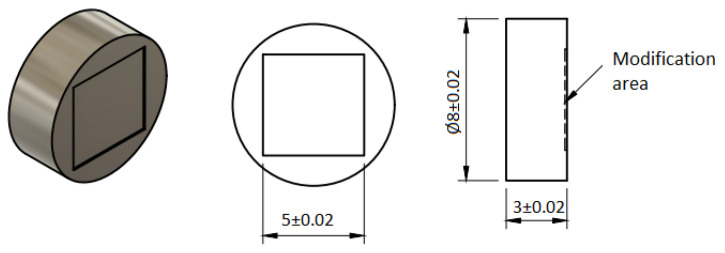
Diagram of the laser-modified sample.

**Figure 2 jfb-14-00478-f002:**
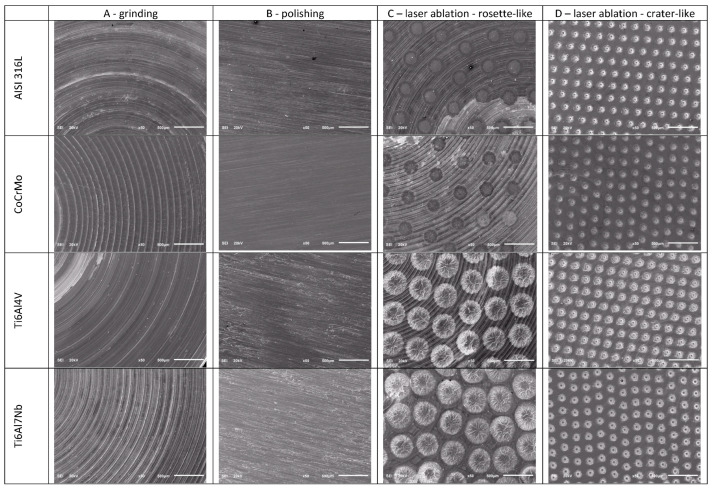
Representative images of the surface of the tested samples at 30× magnification.

**Figure 3 jfb-14-00478-f003:**
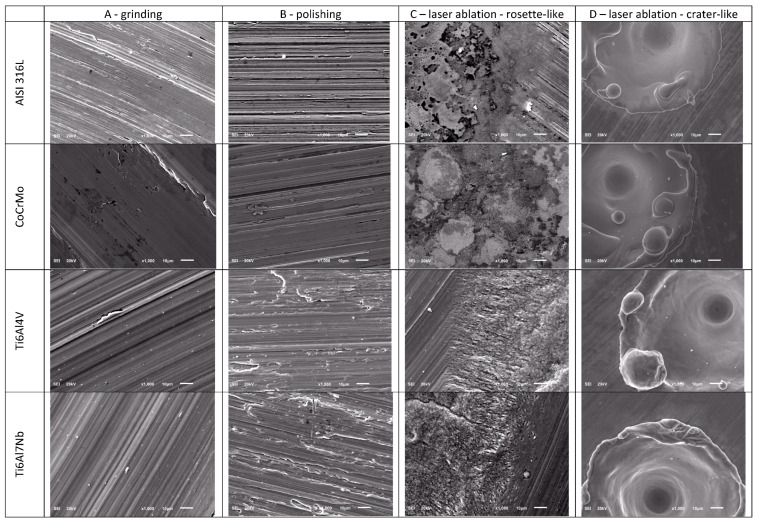
Representative images of the surface of the tested samples at 1000× magnification.

**Figure 4 jfb-14-00478-f004:**
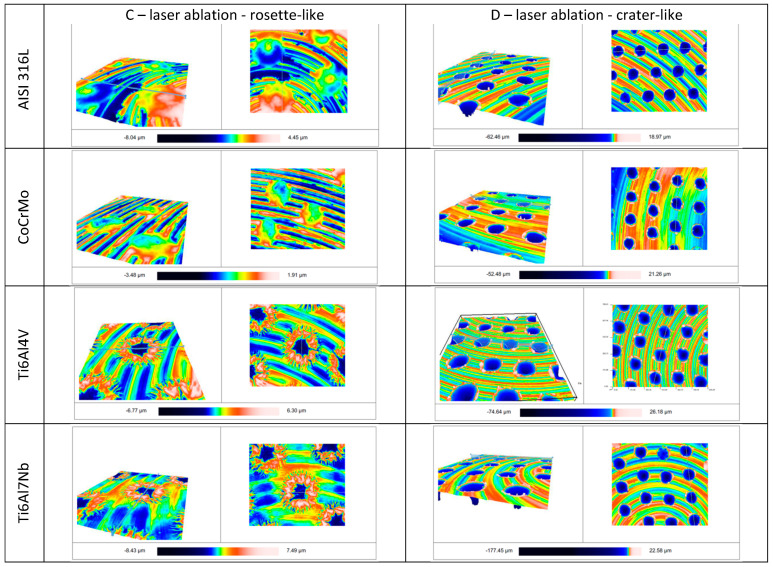
Randomly selected images of the surface of the tested samples C and D obtained with the use of the S neox non-contact optical profilometer. The surface roughness is shown on a color scale.

**Figure 5 jfb-14-00478-f005:**
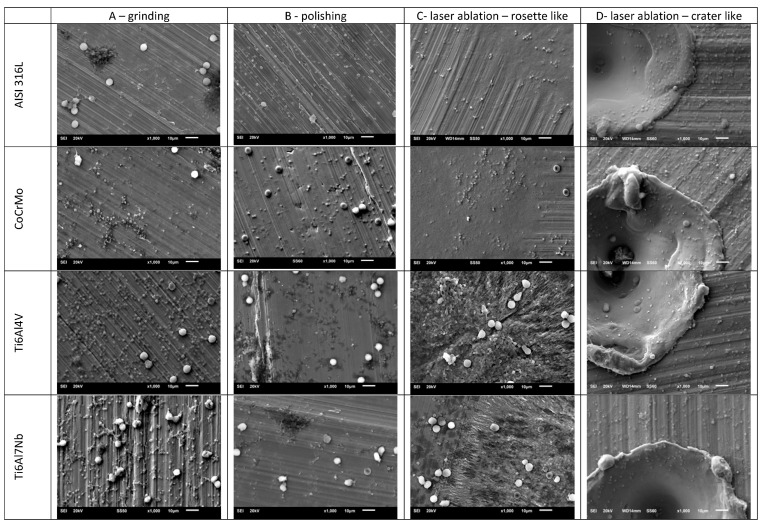
Sample images of the surfaces of the tested materials after contact with blood. Adhered platelets, both single and in the form of aggregates of various sizes, are visible. Numerous leukocytes (large, bright spherical objects) and erythrocytes (large objects with a characteristic concavity) trapped on the surface of the examined materials are also visible.

**Table 1 jfb-14-00478-t001:** Laser Da Vinci 1300 operating parameters.

Sample Modification	Laser Head	Frequency (Hz)	Impulse Duration (ms)	Average Power (W)	Focus Position (mm)	Nozzle-to-Sample Distance (mm)	Spacing (mm)	Gas
C rosette like	FLS352	5	1.2	1000	2.7	1.5	0.50	Air
D crater like	LFS300 OEM	10	10	0.2	3.7	0.5	0.25	Air

**Table 2 jfb-14-00478-t002:** The Ra and Sa parameters determined for the tested samples. The results of the Ra value are presented as the arithmetic mean of five independent measurements and the corresponding standard deviations (MEAN ± SD). The statistical significance of differences in the mean values of Ra calculated for the same biomaterial and its various surface modifications (in rows) and for the same surface modification and various biomaterials (in columns) are presented. The notation “** A–D” means a statistically significant difference at the level of *p* < 0.01 of Ra for the analysed biomaterial subjected to modifications A and D. Whereas the entry “* {1}–{4}” means a statistically significant difference at the level of *p* < 0.05 of Ra for a given modification and biomaterials AISI 316L {1} and Ti6Al7Nb {4}. The notation “ns” means no statistically significant differences within the compared groups. Sa results are presented in the form of averaged height values from the measurement area provided by the SensoSCAN 6 software. Statistical significance levels are marked with asterisks as follows: * *p* < 0.05, ** *p* < 0.01, *** *p* < 0.001, ns—not significant.

Sample	Modification Type								
	Ra (Mean ± SD, µm) *n* = 5					Sa (Mean, µm)			
	A Grinding	B Polishing	C Rosette Like	D Crater Like	Significance	A Grinding	B Polishing	C Rosette Like	D Crater Like
AISI 316L {1}	1.23 ± 1.13	0.34 ± 0.09	0.52 ± 0.17	3.21 ± 0.59	** A–D, C–D *** B–D	1.18	0.40	1.13	4.66
CoCrMo {2}	1.56 ± 0.91	0.32 ± 0.08	0.67 ± 0.07	4.19 ± 0.30	* A–B *** A–D, B–D, C–D	1.43	0.29	0.60	4.16
Ti6Al4V {3}	3.58 ± 1.11	0.45 ± 0.17	1.59 ± 0.21	4.65 ± 1.60	** A–B, C–D *** B–D	1.43	0.41	1.15	7.82
Ti6Al7Nb {4}	1.58 ± 1.38	0.30 ± 0.07	0.90 ± 0.11	4.92 ± 1.26	** A–D *** B–D, C–D	0.77	0.28	1.23	7.77
significance	ns	ns	* {1}–{4} *** {1}–{3}, {2}–{3}, {3}–{4}	ns					

**Table 3 jfb-14-00478-t003:** The number of platelets adhered to the surface of the tested biomaterials, broken down by the type of surface modification and the degree of activation. Numerical results presented as arithmetic means and the corresponding standard deviations (MEAN ± SD) from the number of platelets counted on the tested surfaces of biomaterials. Each of the presented mean values represents the analysis of 120 independent images (except for the column “Mean per biomaterial” where it represents the analysis of 480 images). Explanations of the other markings are in the text above. Statistical significance levels are marked with asterisks as follows: * *p* < 0.05, ** *p* < 0.01, *** *p* < 0.001, ns—not significant.

	Activation Type	Mean per Biomaterial		A Grinding	B Polishing	C Rosette Like	D Crater Like	Significance
AISI 316L {1}	no activation	66 ± 81	8 ± 14	7 ± 7	3 ± 3	4 ± 4	18 ±23	*** A–D, B–D, C–D
	moderate		111 ± 96	84 ±73	120 ± 84	78 ± 47	162 ± 123	** A–B, B–C, B–D *** A–D, C–D
	strong		79 ± 72	95 ± 94	91 ±76	66 ± 33	61 ± 50	* B–C ** B–C *** A–D
CoCrMo {2}	no activation	67 ± 75	8 ± 13	7 ± 5	4 ± 4	3 ± 4	17 ± 21	* A–C *** A–D, B–D, C–D
	moderate		115 ± 86	95 ±73	124 ± 106	97 ± 57	140 ± 88	* A–B *** A–D, C–D
	strong		74 ±52	67 ± 41	87 ± 53	79 ± 50	61 ± 58	ns
Ti6Al4V {3}	no activation	61 ± 70	9 ± 18	7 ± 5	2 ± 2	6 ± 6	21 ± 31	*** A–D, B–D, C–D
	moderate		95 ± 87	66 ± 58	108 ± 84	91 ± 72	124 ± 112	* C–D ** A–B *** A–D
	strong		64 ± 51	70 ± 54	82 ± 54	65 ± 43	49 ± 51	** A–D *** B–D
Ti6Al7Nb {4}	no activation	57 ± 61	11 ± 19	9 ± 7	5 ± 5	6 ± 8	24 ± 34	*** A–D, B–D, C–D
	moderate		88 ± 71	95 ± 73	111 ± 105	84 ± 70	94 ± 82	ns
	strong		61 ± 54	67 ± 41	105 ± 87	50 ± 38	33 ± 37	*** A–B, A–D, B–C, B–D
significance	no activation	ns	* {1}–{4}, {2}–{4}	ns	ns	ns	ns	
	Moderate		* {1}–{3}, ** {2}–{3}, *** {1}–{4}, {2}–{4}	* {2}–{3}	ns	ns	** {1}–{3},{1}–{4}	
	Strong		* {2}–{3} ** {2}–{4} *** {1}–{3}, {1}–{4}	ns	ns	** {2}–{4}	*** {2}–{4}	

**Table 4 jfb-14-00478-t004:** The number of platelets adhered to the surface of the tested biomaterials, broken down by the type of surface modification and the degree of aggregation. Other explanations as in Table 3 and in the text of Section 3.4. Statistical significance levels are marked with asterisks as follows: * *p* < 0.05, ** *p* < 0.01, *** *p* < 0.001, ns—not significant.

	Aggregation Type	Mean per Biomaterial		A Grinding	B Polishing	C Rosette Like	D Crater Like	Significance
AISI 316L {1}	single platelets	42 ± 53	89 ± 62	69 ± 40	78 ± 59	73 ± 38	138 ±76	*** A–D, B–D, C–D
	small aggregates		32 ± 26	17 ± 11	37 ± 23	28 ± 22	45 ± 35	* B–C, ** A–C *** A–B, A–D, C–D
	large aggregates		4 ± 6	6 ± 8	4 ± 5	1 ± 3	3 ± 5	* A–B, C–D *** A–C, A–D, B–C
CoCrMo {2}	single platelets	42 ± 52	88 ± 62	93 ± 69	74 ± 55	74 ± 46	115 ± 67	* A–D *** B–D, C–D
	small aggregates		35 ± 21	23 ± 13	43 ± 23	32 ± 15	38 ± 23	** A–C *** A–B, A–D, C–D
	large aggregates		3 ± 5	2 ± 4	3 ± 4	4 ± 5	3 ± 5	** A–C
Ti6Al4V {3}	single platelets	33 ± 39	68 ± 44	62 ± 35	57 ± 47	66 ± 41	90 ± 46	*** A–D, B–D, C–D
	small aggregates		27 ± 22	19 ± 12	31 ± 21	26 ± 20	33 ± 29	*** A–B, A–D
	large aggregates		3 ± 6	3 ± 4	5 ± 8	2 ± 4	2 ± 4	* A–B *** B–C, B–D
Ti6Al7Nb {4}	single platelets	32 ± 37	65 ± 42	67 ± 37	64 ± 46	61 ± 44	69 ± 40	ns
	small aggregates		27 ± 21	21 ± 16	35 ± 21	22 ± 18	29 ± 23	* A–D, C–D *** A–B, C–B
	large aggregates		4 ± 5	2 ± 4	6 ± 7	2 ± 3	3 ± 4	*** A–B, B–C, B–D
significance	single platelets	** {1}–{3}, {1}–{4}, {2}–{3}, {2}–{4}	* {1}–{3}, {2}–{3} ** {1}–{4}, {2}–{4}	*** {2}–{1}, {2}–{3}, {2}–{4}	** {2}–{3} *** {1}–{3}	ns	** {3}–{4} *** {1}–{2}, {1}–{3}, {2}–{4}	
	small aggregates		* {2}–{3}, {2}–{4}	ns	ns	ns	ns	
	large aggregates		ns	ns	ns	ns	ns	

**Table 5 jfb-14-00478-t005:** Spontaneous platelet activation in citrated whole blood, assessed by expression of P-selectin (CD62P), after contact with the tested biomaterials. Evaluation results are presented as fluorescence intensity (a.u.) and represent the mean values and standard deviations (MEAN ± SD) of four independent experiments with three replicates of each sample (*n* = 12). For negative and positive controls *n* = 48. Explanations of other symbols are in the text. Statistical significance levels are marked with asterisks as follows: * *p* < 0.05, *** *p* < 0.001, ns—not significant.

	A Grinding	B Polishing	C Rosette Like	D Crater Like	Significance
AISI 316L {1}	308 ± 161	348 ± 190	470 ± 236	258 ± 87	* C–D
CoCrMo {2}	322 ± 137	250 ± 87	443 ± 271	236 ± 80	* C–D, *** A–D, B–D
Ti6Al4V {3}	345 ± 210	246 ± 61	410 ± 189	286 ± 121	ns
Ti6Al7Nb {4}	327 ± 171	246 ± 62	421 ± 199	281 ± 76	* B–C
negative control {5}	350 ± 173				*** {5}–{6}
positive control {6}	534 ± 429				
significance	ns	ns	ns	ns	

**Table 6 jfb-14-00478-t006:** Spontaneous platelet aggregation in citrated whole blood after contact with the tested biomaterials assessed by scattering laser light forward and sideways (a.u.). The results shown represent the mean values and standard deviations (MEAN ± SD) of the percentage of aggregated platelets from four independent experiments with three replicates of each sample (*n* = 12). For negative and positive controls *n* = 48. Explanations of other symbols are in the text. Statistical significance levels are marked with asterisks as follows: *** *p* < 0.001, ns—not significant.

	A Grinding	B Polishing	C Rosette Like	D Crater Like	Significance
AISI 316L {1}	1.14 ± 0.25	1.24 ± 0.26	1.53 ± 0.57	1.12 ± 0.37	ns
CoCrMo {2}	1.29 ± 0.40	1.30 ± 0.18	1.68 ± 0.46	1.32 ± 0.59	ns
Ti6Al4V {3}	1.31 ± 0.39	1.16 ± 0.19	1.49 ± 0.52	1.18 ± 0.44	ns
Ti6Al7Nb {4}	1.28 ± 0.34	1.29 ± 0.21	1.64 ± 0.65	1.23 ± 0.57	ns
negative control {5}	1.34 ± 0.44				*** {5}–{6}
positive control {6}	1.88 ± 0.77				
significance	ns	ns	ns	ns	

## Data Availability

The raw/processed data required to reproduce these findings cannot be shared at this time as the data also form part of an ongoing study.

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
