# Peer review of "Adhesion and Activation of Blood Platelets on Laser-Structured Surfaces of Biomedical Metal Alloys"

_jfb, 2023, doi:10.3390/jfb14090478_

Round 1
Reviewer 1 Report
The manuscript presents a comparative study of different surface structures on adhesion and activation of blood platelets. The work is interesting, but following points may be addressed to improve the quality and readability of the manuscript-
1. The dimensions (depth, spacing etc.) of the laser-processed structures should be mentioned.
2. The process of determining activation level should be clearly mentioned.
3. The results should be reported with the help of graphs to help readers in understanding.
4. Relevant recent works on long-pulsed laser texturing (e.g. https://doi.org/10.1080/10667857.2017.1390931) may be consulted for discussing the laser processing process.
5. Is there any pattern that performs better than all the others for all types of materials? Similarly, is there any material that performs better than all the others for all types of surface textures?
6. The authors have mentioned that the laser processing was carried out in air, which may induce nitridation and oxidation. It would help if the authors present surface chemical composition before and after the processing.
The quality of the language of the manuscript is acceptable.
Reviewer 2 Report
Authors present the Laser-structured surfaces of biomedical metal alloys for Adhesion and activation of blood platelets that within the Journal of Functional Biomaterials. In order to increase the readability of the paper, some amendments are suggested as follows:
A). A quantified numerical value of the abstract is necessary.
B). Basis for choosing laser (such as wavelength and processing conditions, etc.).
C). Patterned Adhesion and activation of blood platelets experimental data information should be presented in the manuscript.
Minor editing of English language required
Reviewer 3 Report
The current manuscript introduces a captivating and significant topic in the field of Implantology. The authors have undoubtedly put forth commendable effort; however, prior to reaching a final decision, several areas require careful revision. Foremost, there is a need for the authors to enhance the clarity and coherence of the lengthy sentences by employing academic English and taking into account the provided comments for guidance:
- Commencing with the title, it is essential to ascertain the implications of "laser-structured surfaces…"; are these surfaces primarily focused on decontamination, or is there an alternate aim? It might be prudent to consider refining the title in the revised version, aligning it more closely with the work's objectives for enhanced clarity.
- The introduction, along with its aim, requires refinement. The aim is overly lengthy and lacks clarity. Additionally, the authors have indicated their aim in line 48; thus, it is advisable to elucidate the introduction prior to outlining the work's objectives. Introducing a hypothesis subsequent to the aim could enhance the text, while also providing justification for the chosen groups. Notably, lines 84-87 appear to be misplaced, as they pertain to the methodology description.
- Moreover, the referee suggests considering the following articles for incorporation into the introduction section: [Int J Environ Res Public Health. 2021 Apr 27;18(9):4670. doi: 10.3390/ijerph18094670. PMID: 33925742].
- In the "Materials and Methodology" section, kindly provide the complete names of the metal alloys prior to their acronyms, especially during their initial mention. Similarly, ensure the full names are given before abbreviations such as PBS and ADP in line 97.
- The content of subsection 2.3 appears to create confusion. The process of preparation lacks clarity. Employing a flowchart, scheme, or experimental design might alleviate this confusion for readers. Additionally, the manufacturer of the utilized laser is not specified in the text. The nature of the "rosette-like effect and crater-like effect" warrants clarification; are these effects inherent to the laser or were the materials initially structured in this manner? A thorough and clear revision of the methodology is necessary.
- In line 119, please include the acronym SEM after the complete name and subsequently omit it in the subsequent sections.
- Could you kindly provide the city in Germany where JENAOPTIK is located? Furthermore, in reference to the EVOVIS 2.00.1.00 software, consider including additional information such as the year of production or the software's producer.
- For line 131, include the full name of "Ra" prior to the acronym.
- It is advisable to reposition subsection 2.6 as the initial one and rename it as 2.1 for improved clarity in the structure.
- Maintain consistency in expressions for time units. For instance, in line 149, "1 hour" is used, while in line 151, "one hour" is present. Ensure uniformity throughout the document by using the same expression.
- Better to express in lowercase the word "absolute" in line 153.
- Lines 154-156 appear to be a component of the results section, and the same applies to lines 156-158.
- Additionally, lines 159-165 leave readers puzzled. The nature of the procedure for platelet division isn't evident—could you confirm whether this pertains to a methodology or a scale applied in the results?
- The inclusion of the phrase "for the purpose of the experiment" in line 176 requires further context to comprehend its relevance.
- Could you clarify if CellFix reagent is produced by Becton or if it is associated with a different manufacturer?
- In the "Results" section, there is a misidentification where figures are referred to as tables, specifically in 3.1 where figures 2 and 3 are presented. This discrepancy needs correction, along with the creation of new figure legends to accompany these figures.
- Further clarification is needed regarding the HOMMEL TESTER T1000. Could you provide the manufacturer's name for this equipment?
- The title and description of table 4 (lines 206-218) appear to be causing confusion. This should be improved.
- Similarly, "table 5" should be revised to "figure 5." The term "Exemplary" might not be suitable in a scientific context. Please consider finding a more appropriate term to describe the contents of figure 5 and add a legend to enhance the understanding of the images within the figure.
- In line 225, replace the full name with "SEM." Additionally, rephrase the sentence starting with "number 120" in line 226 for improved clarity.
- The same applies to "table 6," which should be "figure 6." Instead of "photographs," maintain consistent terminology, such as "images," to match the previous figure descriptions. Be sure to include a comprehensive legend for figure 6.
- In line 238, could you specify the section number that corresponds to the described groups?
Place "table 7" within subsection 3.4 and "table 8" within subsection 3.5. Additionally, elaborate on the descriptions of these respective subsections to enhance clarity. The same adjustments should be made for tables 9 and 10, which currently lack clear descriptions and create confusion within their respective sections (3.6 & 3.7).
- In the "Discussion" section, it's noted in line 74 that the authors refer to microscopic analysis as "tables," but these are actually images. Therefore, it is recommended to replace "tables" with "figures" for accuracy.
- Regarding the terms "photographs" and "images," it's important to maintain consistent terminology throughout the manuscript. While these terms might be interchangeable to some extent, for the sake of clarity and uniformity, it's advisable to choose one term and use it consistently throughout the text.
- Following the revisions made in the previous sections, it is crucial for the authors to thoroughly review and recheck the "Discussion" section in light of these changes. This will ensure that the discussion remains coherent and aligns with the adjusted content.
- In the "Conclusions" section, the initial sentence should be refined and presented in a more concise, scientific/academic manner. The current lengthy sentence (lines 165-172) is not clear and can be confusing in its meaning. Additionally, the term "report" used by the authors to describe this work and the phrase "very subtle..." (line 165) need further clarification.
- Kindly ensure that the “references” are corrected according to the journal's guidelines for formatting and citation.
The authors have undertaken commendable efforts, but there is space for improvement in the manuscript's quality as per the suggested comments. Additionally, a thorough review of the English language is required, as there are sentences that currently lack clarity and are difficult to comprehend.
Round 2
Reviewer 2 Report
Author's has been revise and correct reviewer's comment, i suggestion this manuscript can be accept for publication in this status.
Minor editing of English language required
Reviewer 3 Report
The authors have enhanced their manuscript in response to the provided comments!
The English is better!